# Threatening Facial Expressions Impact Goal-Directed Actions Only if Task-Relevant

**DOI:** 10.3390/brainsci10110794

**Published:** 2020-10-29

**Authors:** Christian Mancini, Luca Falciati, Claudio Maioli, Giovanni Mirabella

**Affiliations:** 1Department of Clinical and Experimental Sciences, University of Brescia, Viale Europa 11, 25123 Brescia (BS), Italy; christian.mancini@unibs.it (C.M.); luca.falciati@unibs.it (L.F.); claudio.maioli@unibs.it (C.M.); 2IRCCS Neuromed, Via Atinense 18, 86077 Pozzilli (IS), Italy

**Keywords:** motor readiness, emotion, facial expressions, decision making, Go/No-go task

## Abstract

Facial emotional expressions are a salient source of information for nonverbal social interactions. However, their impact on action planning and execution is highly controversial. In this vein, the effect of the two threatening facial expressions, i.e., angry and fearful faces, is still unclear. Frequently, fear and anger are used interchangeably as negative emotions. However, they convey different social signals. Unlike fear, anger indicates a direct threat toward the observer. To provide new evidence on this issue, we exploited a novel design based on two versions of a Go/No-go task. In the emotional version, healthy participants had to perform the same movement for pictures of fearful, angry, or happy faces and withhold it when neutral expressions were presented. The same pictures were shown in the control version, but participants had to move or suppress the movement, according to the actor’s gender. This experimental design allows us to test task relevance’s impact on emotional stimuli without conflating movement planning with target detection and task switching. We found that the emotional content of faces interferes with actions only when task-relevant, i.e., the effect of emotions is context-dependent. We also showed that angry faces qualitatively had the same effect as fearful faces, i.e., both negative emotions decreased response readiness with respect to happy expressions. However, anger has a much greater impact than fear, as it increases both the rates of mistakes and the time of movement execution. We interpreted these results, suggesting that participants have to exploit more cognitive resources to appraise threatening than positive facial expressions, and angry than fearful faces before acting.

## 1. Introduction

Social cognition, i.e., ability to make sense of others’ behavior, intentions, and emotions, is aimed at achieving a mutual understanding between individuals, allowing each to produce adaptive actions within a given context. In this sense, social cognition is a particular category of decision-making processes. The recognition of emotional facial expressions profoundly influences this process, as it automatically triggers appropriate behaviors in a social environment [1,2]. For instance, the sight of angry or fearful faces of conspecifics automatically triggers defensive responses in the observer [3,4]. Conversely, a happy facial expression promotes approaching behaviors [5]. Impairments in recognition of facial emotions such as those occurring in psychopathy [6] or schizophrenia [7], or after bilateral amygdala lesions [8], severely compromise social interactions. 

Despite the undeniable link between emotional and motor processes, the empirical evidence about how facial emotional expressions influence action preparation or response inhibition is highly contradictory. For instance, Berkman et al. [9], exploiting a Go/No-go task, did not find differences in reaction times (RTs) or accuracy during Go-trials between positive and negative images of faces. Similarly, Schulz et al. [10] found that sad, happy, and neutral faces did not affect response readiness. However, the same authors in a previous paper [11], using a similar Go/No-go task, found that happy faces elicited faster RTs and more errors than sad faces. Finally, Zhang and Lu [12] found decreased RTs and higher accuracy for positive and negative images of faces with respect to neutral facial images in an emotional Go/No-go task.

The impact of emotional faces on motor response inhibition, which is a key executive function for motor control [13], is also unclear. Exploiting the stop-signal task [14], Sagapse et al. [15] found that the stop-signal reaction time (SSRT), i.e., the estimated time it takes to cancel a planned movement, was not affected by the presentation of fearful faces with respect to neutral faces. However, Rebetez et al. [16] showed that both happy and angry faces increased the length of the SSRT with respect to neutral faces. By contrast, Derntl and Habel [17] reported that angry faces elicit faster SSRTs with respect to neutral faces in schizophrenic and age-matched participants.

Differences in experimental designs can only partially account for such conflicting results. First, response modulations elicited by stimuli with negative valence such as fearful faces are different from those elicited by other negative valence stimuli such as angry or sad faces because they communicate diverse messages [18]. Second, stimulus arousal is often not considered, even though this dimension of emotional stimuli has an impact on response modulation [19]. However, in our view, the factor that mostly affects previous results is the task-relevance of emotional stimuli. In most studies, the emotional content of the stimuli was irrelevant with respect to the task instructions. Under these conditions, the influences of emotions on motor behavior are highly subjective and variable. Furthermore, in the cases in which the emotional valence of facial images was used as a cue for motor responses in a Go/No-go task [10,11,20,21,22], participants were required to move on one emotional facial expression (e.g., happy faces) and to withhold their actions on a different emotional expression (e.g., angry faces) and vice versa. Therefore, since two different responses had to be performed according to the stimulus valence, the modulation of action readiness was conflated with task switching. Importantly, these studies lack a control condition in which emotional faces, i.e., stimuli with the same visual features, are presented, but are task-irrelevant [10,11,20,21,22].

To overcome these limitations, Mirabella [23] devised an experimental design in which participants had to perform two versions of a Go/No-go task. In the emotional Go/No-go paradigm, participants had to perform the same movement when pictures of emotional facial expressions with different valence (fear or happy) but matched for arousal were presented. In contrast, in the gender Go/No-go task, the same pictures were shown, but participants had to move according to the actor’s gender, disregarding the emotional valence of the face. Results showed that, in the emotional task, RTs increased and accuracy worsened when people responded to fearful with respect to happy faces. Intriguingly, all effects disappeared in the gender task [23]. For the first time, this approach allowed two different facial emotional valences to be compared on the planning of the same action according to their task-relevance. This evidence indicated that, when task relevant, fearful expressions capture attention more than happy faces, possibly to detect the source of potential threats. One open question is how threatening facial expressions, i.e., angry and fearful face, modulate actions. In fact, those expressions convey different social signals. Unlike fear, anger conveys a direct threat toward the observer, eliciting immediate action. Thus, it is likely they could elicit different responses. Even though a few studies showed that anger affects action planning differently with respect to fear [18,24,25,26], further studies are needed. To this end, we replicated the Mirabella [23] experiment, adding a third facial emotion depicting anger. We aim to assess whether (i) the effect of the angry facial expression would depend on task-relevance, (ii) angry faces speed up the motor response with respect to fearful faces (given that they could represent a direct threat), and (iii) people remember differentially emotional facial expressions.

## 2. Materials and Methods

### 2.1. Participants

All subjects recruited for the study (56 participants, 28 males and 28 females, mean ± SD age: 22.36 ± 2.41) were right-handed, as assessed with the Edinburgh handedness inventory [27] and had a normal or corrected-to-normal vision. None of the participants had a history of neurological or psychiatric disorder and were naïve about the purpose of the study. The study was conducted in accordance with the ethical guidelines set forth by the Declaration of Helsinki and approved by the institutional review board of the Istituto di Ricovero e Cura a Carattere Scientifico (IRCCS) Neuromed Hospital, Pozzilli (IS) Italy (Prot. 14/18). Informed consent was obtained from all participants. Data will be freely available from the Open Science Framework platform [28].

### 2.2. Stimuli

A total of 16 different grayscale pictures of faces (two males and two females) were selected from the Pictures of Facial Affect [29] and used as a stimuli. Each actor displayed four different facial expressions (fear, happiness, anger, and neutral). At the end of the experimental session, after the emotion recall (see below), participants were asked to fill out a questionnaire to evaluate the level of arousal, valence, and recognizability of each image. An 8-point Likert scale assessed arousal (0 meant ‘no arousing’ and 7 meant ‘highest arousing’). A 15-points Likert scale assessed valence (−7 meant ‘very negative,’ 0 meant ‘neutral,’ +7 ‘very positive’). The recognizability of emotions was assessed as described in the handout of the Pictures of Facial Affect [29]. Briefly, all images were shown again to participants, and, for each picture, they had to score the recognizability of fear, happiness, and anger on an 8-points scale (0 meant ‘no emotion was recognized’ and 7 meant ‘emotion was fully recognized’). Thus, each image ended up having three scores, indicating how much the picture expressed each of the three emotions.

First, post hoc tests on arousal (pairwise comparisons with Bonferroni correction) showed that (i) anger had significantly lower scores than happiness and fear, and (ii) fear had a lower value than happiness (Table 1, Figure 1A).

Second, post hoc tests on valence revealed that (i) faces expressing fear were evaluated more negatively than those expressing anger, (ii) faces expressing fear and anger were evaluated more negatively than those expressing happiness, as expected (Table 1, Figure 1B).

Finally, post hoc tests on emotional recognition showed that angry faces were judged less recognizable than happy and fearful faces, meaning that the facial expressions of fear and happiness were recognized more easily than anger (Table 1, Figure 1C). Furthermore, happiness was more recognizable than fear, which is in line with the literature [29,30].

### 2.3. Experimental Apparatus and Behavioral Tasks

Participants sat in a dimly illuminated, quiet room. Visual stimuli were displayed on a 17-inch Liquid Crystal Display touchscreen monitor (MicroTouch M1700SS, 3M, Minnesota, MN, USA, 1280 × 1024 resolution, 32-bit color depth, refresh rate 75 Hz, sampling rate 200 Hz) against a black background of uniform luminance (<0.01 cd/m^2^). The monitor was about 40 cm from the eyes of the participants. Images all had the same dimension (5.8 cm × 7.4 cm or 8.25 × 10.9 degrees of visual angles, dva). The temporal arrangements of stimulus presentation were synchronized with the monitor refresh rate (75 Hz). CORTEX, which is a non-commercial software package, was used to control both stimulus presentation and behavioral responses [31]. As in Mirabella [23], participants had to perform two versions of a Go/No-go task (the Emotion and the Gender discrimination task) in a single experimental session. The order of administration of the tasks was counterbalanced across participants. A ten-minute break was interposed between the execution of the two tasks.

### 2.4. Emotion Discrimination Task

Each trial started with a visual stimulus presentation consisting of a central red circle (2.43 cd/m^2^, diameter 2.8 cm or 4 dva) positioned 2 cm below the center of the screen. Participants had to reach it with their right index finger (Figure 2A). As soon as the central stimulus was touched, a peripheral red circle (diameter 2.8 cm or 4 dva) appeared to the right of the central stimulus at an eccentricity of 8 cm or 11.3 dva. Participants had to hold the central stimulus for 400–700 ms. After this period, the central stimulus disappeared and, simultaneously, one of the four pictures depicting a facial expression appeared just above the tip of the index finger. Whenever an emotional face was presented, subjects were instructed to lift the index finger and touch the peripheral target as quickly and accurately as possible, holding it for 300–400 ms. Conversely, whenever a neutral facial expression was shown, participants had to refrain from moving by keeping the finger on the central position for 400–800 ms. We never told participants what emotions would be presented. The instructions were simply to move whenever they recognized an emotion, and to refrain from moving whenever they saw an emotionless facial expression. Acoustic feedback indicated successful trials, i.e., when subjects correctly moved their index finger in Go-trials, and when they correctly withhold their movements in No-go trials.

Each picture showing an emotional face was presented 22 times (Go-trials, frequency: 67%) whereas each neutral face was presented 33 times (No-go trials, frequency: 33%). Images were presented in a pseudorandom order. Overall, participants had to perform 396 trials. The task consisted of two blocks with a resting period allowed between blocks or whenever requested. Pictures were randomly intermixed within the task. Participants were discouraged from slowing down during the task by setting an upper-RT limit for Go-trials, i.e., every time RTs were longer than 500 ms, the Go-trials were considered as errors. However, to avoid cutting the RT distribution’s right tail, participants had an additional time of 100 ms for releasing the central stimulus (overtime reaching-trials, see Reference [32]). Thus, every time participants detached the index finger after 500 ms, but before 600 ms, the RT was recorded, i.e., participants were allowed to detach the finger, but then, all images disappeared, signaling an error. Overtime reaching-trials accounted for 6.2% of the total Go-trials and were included in the analyses. Finally, every time RTs were longer than 600 ms, trials were aborted, i.e., all images were turned off, signaling an error.

### 2.5. Gender Discrimination Task

The Gender discrimination task had the same time course and the same stimuli as the Emotion discrimination task, but participants had to move according to the gender of the face (Figure 2B). To avoid any gender bias, half of the participants had to move when male faces were presented, and to withhold the movement in the presence of female faces (male-control condition), the other half of participants had to perform the task the other way around (female-control condition). In the male-control condition, each set of pictures showing one male actor face was presented 132 times (Go-trials, frequency: 67%), whereas each set of pictures showing one female actor face was presented 66 times (No-go trials, frequency: 33%), and vice versa for the female control condition. Images were presented in a pseudo-random order. Overall, participants had to perform 396 trials, as we had two male and two female actors. Overtime reaching-trials accounted for 2.8% of the total Go-trials and were included in the analyses.

### 2.6. Emotion Recall

At the end of the experimental session and before the stimuli rating, participants were given a surprise recognition task in which they were asked to remember what emotions were shown in the pictures. Since the order of administration of the two tasks was counterbalanced across subjects, one half of the participants recalled the emotions after the Emotion discrimination task (emotion recall group) and the other half after the Gender discrimination task (gender recall group). Participants were allowed to report the emotions for a period of two minutes. Importantly, the order of emotion recall was carefully written down. A smaller percentage of participants (12.5%) recalled only two facial emotional expressions. The other 87.5% recalled at least three emotions (overall, about 9% of participants recalled four emotions). Given that only a small percentage of participants remembered four emotions, we considered only the first three recalled emotions.

### 2.7. Data Analyses

RTs, movement times (MTs) of correct Go-trials, and errors in Go-trials were taken as behavioral parameters. RTs were determined as the time difference between the Go-signal presentation and the movement onset, i.e., the instant when the finger was detached from the screen. MTs were computed as the time interval between the movement onset and the moment in which the peripheral target was touched. All Go-trials that had RTs longer than the mean plus three SDs, and those shorter than the mean minus three SDs were excluded from the analysis. Therefore, a total of 0.74% of the data was eliminated. Errors in Go-trials were defined as those instances in which participants, instead of reaching the peripheral target, kept their index finger on the central stimulus. The error rate was computed for each participant and determined as the ratio between the number of errors in a given condition (e.g., anger in the Emotion discrimination task), and the overall number of trials for the same condition (e.g., all Go-trials in which an angry face was shown). We averaged the arousal scores for angry, fearful, and happy faces to compute the overall arousal level (AL) for each participant. To use this measure as an independent variable in the statistical analyses, we ranked participants according to the AL values, and we subdivided them, without setting thresholds, into three groups (high, medium, and low AL). Each group was composed of 18, 20, and 18 subjects, respectively.

Three different four-way ANOVAs with a mixed design [between-subjects factor: AL (high, medium, low) and Sex (male, female), within-subject factors: Emotion (anger, fear, happiness) and Task (Emotion discrimination task, Gender discrimination task)] were performed to analyze RTs, MTs, and error rates across experimental conditions. Bonferroni corrections were applied to all post hoc tests. All analyses were performed on the average values of each behavioral parameter.

We provided a measure of the effect size. To this end, we calculated the partial eta-squared (η^2^_p_, values equal or above 0.139, 0.058, and 0.01 indicating large, medium, and small effects, respectively) for each ANOVA and the Cohen’s *d* (values equal or above 0.8, 0.5, and 0.2 indicating large, medium, and small effects, respectively) as the effect size for each *t*-test [33]. Finally, to quantify the strength of the null hypothesis, we calculated the Bayes factors (BF_10_) with an r-scale of 0.707 [34]. BF_10_ values < 0.1 and <0.33 provide strong and moderate support, respectively, for a null hypothesis. Conversely, BF_10_ values >3 and >10 constitutes moderate and strong support for the alternative hypothesis. To improve readability, in the following, we report only significant results, unless otherwise indicated. However, all statistical results are reported in Appendix A.

In addition, via two Chi-Square Goodness-of-fit tests, we compared (i) the overall percentage of recalled emotions, and (ii) the percentage of the first, second, and third recalled emotions in the emotion and gender recall group against the chance level. To identify which cells from the contingency table differed from their expected values when the Chi-squared tests showed significant results, we used the standardized cell residuals (StdR). The StdR are derived by raw residuals, which are computed as the difference between the observed value and the expected value divided by the expected value’s squared root [35], using the following formula.
(1)$$StdR=\frac{x_{ij}-\;e_{ij}}{\sqrt{e_{ij}}}$$
where *e_ij_* is the expected value of the *ij*-th cell. The computed values of the StdR were compared to the critical value from the statistical parameter distribution using the R-Stats-package [36]. Bonferroni corrections were applied to all post hoc tests. Following the same procedure, we also compared (i) the overall percentage of recalled emotions, and (ii) the percentage of the first, second, and third recalled emotions across the two tasks via Chi-square tests of Independence. Given that no significant results were found; these are not further discussed.

## 3. Results

### 3.1. Analyses of RTs

We assessed whether and how emotional faces affect response readiness. The four-way ANOVA on mean RTs of Go-trials revealed a main effect of Emotion, Task, and an interaction between these two factors (Table 2, Figure 3A). The main effect of Emotion was because participants reacted more slowly after the presentation of angry faces (M = 382.8 ms, 95% CI [375.6, 390.0]) than after the presentation of fearful faces (M = 367.5 ms, 95% CI (361.6, 373.3)) and happy faces (M = 362.9 ms, 95% CI (357.3, 368.5)). The main effect of Task indicated that participants reacted faster during the Gender discrimination task (M = 365.7 ms, 95% CI (361.3, 370.2)) than during the Emotion discrimination task (M = 376.4 ms, 95% CI (370.6, 382.2)). Their interaction qualifies both main effects. During the Emotion discrimination task, the presentation of an angry face significantly increased the RTs (M = 398.7 ms, 95% CI (387.8, 409.5)) with respect to both happy faces and fearful faces (respectively, M = 360.4 ms, 95% CI (352.1, 368.8) and M = 370.0 ms, 95% CI (361.5, 378.4)). Furthermore, in the emotional task, the RTs were longer for fearful than for happy faces, which is in line with the results of Mirabella [23]. In contrast, in the Gender discrimination task, RTs did not differ across the three emotions. Finally, RTs after the presentation of angry faces in the Emotion discrimination task were slower than those occurring for angry faces during the Gender discrimination task (M = 366.9 ms, 95% CI (359.1, 374.7)).

### 3.2. Analyses of MTs

Using the four-way ANOVA design, we also evaluated the effects of emotional faces on mean MTs. This analysis revealed the main effect of Emotion because participants had longer MTs after the presentation of angry faces (M = 323.6 ms, 95% CI (307.2, 339.9)) than after fearful and happy faces presentations (respectively, M = 315.3 ms, 95% CI (300.2, 330.4) and M = 315.9 ms, 95% CI (301.0, 330.9), Table 3, Figure 3B). We also found a significant interaction between Task and Emotion. The interaction was because, in the Emotion discrimination task, MTs after angry faces (M = 331.7 ms, 95% CI (306.0, 357.4)) were longer than those after fearful and happy faces (respectively, M = 316.0 ms, 95% CI (293.1, 339.0) and M = 317.2 ms, 95% CI (293.8, 340)). Furthermore, MTs after angry faces in the Emotion discrimination task were longer than those in the Gender task (M = 315.4 ms, 95% CI (294.7, 336.1)). All these differences can be better appreciated by looking at the mean differences of MTs between each pair of emotional facial expressions in each of the two tasks (Figure 3C).

### 3.3. Analyses of Average Rates of Mistakes

Finally, the four-way ANOVA design was employed to analyze the average rates of mistakes. Four main effects were found (Table 4). The main effect of Emotion (Figure 3D) was because subjects made more mistakes after the presentation of angry faces (M = 7.16%, 95% CI (5.88, 8.43)) than after the presentation of fearful or happy faces (respectively, M = 3.85%, 95% CI (3.06, 4.64) and M = 3.72%, 95% CI (2.99, 4.45)). A higher percentage of mistakes during the Emotion discrimination task (M = 6.90%, 95% CI (5.98, 7.82)) than during the Gender discrimination task (M = 2.92%, 95% CI (2.35, 3.48)) determined the main effect of the Task (Figure 3D). The main effect of Sex (Figure 3E) was explained by the fact that males made more mistakes than females (respectively, M = 5.48%, 95% CI (4.59, 6.37) and M = 4.34%, 95% CI (3.61, 5.07)). Finally, the main effect of the arousal level (AL, Figure 3E) was determined by a higher percentage of mistakes made by participants in the high AL than those in the low AL group (respectively, M = 5.67%, 95% CI (4.52, 8.45) and M = 4.08%, 95% CI (3.22, 4.94)). The main effects of Emotion and Task are qualified by their interactions. Post hoc tests (pairwise comparisons) showed that, in the Emotion discrimination task, the percentage of mistakes occurring after the presentation of angry faces (M = 11.74%, 95% CI (10.05, 13.43)) was higher than after the presentation of fearful and happy faces (respectively, M = 4.67%, 95% CI (3.53, 5.81) and M = 4.28%, 95% CI (3.2, 5.36)). In addition, the rates of mistakes after the presentation of angry and fearful faces was significantly higher in the Emotion discrimination task than in the Gender discrimination task (respectively, M = 2.57%, 95% CI (1.68, 3.46) and M = 3.02%, 95% CI (1.94, 4.11), Table 4). The main effects of Sex and AL are also qualified by their interactions. Males with high AL showed a higher rate of mistakes (M = 8.90%, 95% CI (6.65, 11.15)) than males with medium and low AL (respectively, M = 4.59%, 95% CI (3.28, 5.90) and M = 4. 49%, 95% CI (3.23, 5.74)) and females with high AL (M = 4.06%, 95% CI (2.88, 5.24), Table 4). Importantly, neither Sex nor AL interacted with the factors Task and Emotion.

### 3.4. Correlations between Behavioral Measures and the Recognition Scores

As suggested, emotions recognizability impacts cognitive performance [37]. Therefore, we checked whether behavioral parameters characterizing the Emotion discrimination task’s performance correlates with the recognition scores. To this end, we computed the values of Spearman’s correlation coefficient (rho) between the recognition score of a given emotion and each corresponding behavioral parameter (RT, MT, and error rates). In none of the instances, we found a significant correlation (see Appendix A).

### 3.5. Emotion Recall Performance

The overall frequency of recalled emotions was not significantly different from the chance after both the Emotion (Figure 4A, Table 5) and the Gender discrimination tasks (Figure 4C). However, frequencies of the first recalled emotion were significantly different from the chance level in both groups. In both the emotion and the gender recall group, happiness was remembered more frequently than the chance level. Furthermore, in the gender recall group, anger and fear were recalled less frequently than the chance level. As far as the second recalled emotion, the performance was not different from chance in any of the groups. Finally, only in the emotion recall group, anger was remembered more frequently as a third recalled emotion (Figure 4B,D).

## 4. Discussion

Studies comparing behavioral responses to emotional facial expressions have provided highly contradictory results, which can be explained by three critical factors. First, not infrequently, different types of negative emotions have been used interchangeably [20,21], e.g., facial expressions of anger and fear are often generically considered as threatening stimuli [4,20,21]. However, their social meaning is different and, thus, they are likely to elicit different responses [18]. Second, the arousal of the stimuli is still rarely considered [19], even though it has been shown to influence the experimental outcomes. Third, the task-relevance of emotional stimuli is not considered in most studies, even though it has been shown to play a crucial role [23,37]. In the present research, we addressed all the above key issues. Expanding previous results [23], we assessed the effect of three emotional facial expressions on response planning and execution, considering both their arousal and task-relevance. Our evidence indicates that valence but not arousal or recognizability of the stimuli affects the generation and execution of actions when emotional facial expressions are task-relevant, i.e., during the Emotion discrimination task (see Appendix A). Thus, we confirm and expand our results by showing that the response to not only happy and fearful facial expressions, but also to angry faces, are sensitive to task-relevance. Relevantly, we confirmed that, in the Emotion discrimination task, fearful facial expressions slow down movement preparation more than faces expressing happiness [23]. Importantly, we also showed that angry faces increased RTs, MTs, and the rate of mistakes to a larger extent with respect to fearful faces. Given that we compared the responses elicited by the same stimuli across the same population in the two tasks, they are unlikely to depend on the subjects’ variability or the stimuli features.

### 4.1. Task-Relevance Matters

The present study indicates that emotional stimuli modulate participants’ behavior only when they are task-relevant. Berger et al. [37] obtained similar results, exploiting two versions of a working memory task. A sequence of actors’ faces of different ages and showing neutral, happy, or angry expressions were presented to healthy participants. They were instructed to make match/non-match judgments of the emotion or the age of the actors’ faces with respect to images displaying one or two positions back in the sequence. The authors found that positive emotions facilitated the working memory performance, as judgments were more accurate, and RTs were faster for happy than for neutral and angry faces. However, this effect occurred only when participants were explicitly instructed to respond to emotion. Conversely, when responding to the age of the face, i.e., when emotions were task-irrelevant, the performance for emotional and neutral faces was similar [37]. These results are fully in line with those of Mirabella [23] and those of the present paper. However, this paper extends previous results since it includes the emotion of anger, showing that the effect of task-relevance is not limited to fear and happiness. The null effect of emotions in the Gender discrimination task cannot be ascribed either to stimuli features, as the same stimuli were presented in the Emotion discrimination task, or to interindividual variability as the same persons performed both tasks. Furthermore, the computation of Bayes factors allows us to state that these findings are unlikely due to the variability of the sample or statistical underpowering. Our data suggest that task-irrelevant emotions are unlikely to affect action readiness and execution systematically, at least at a behavioral level. These results could explain contradictory evidence of facial emotional expressions on actions found in previous research. Single subjects can show an impact of emotion, even though they are task-irrelevant. Therefore, especially if the sample is relatively small (less that 15–20 participants), a random effect could be observed. Otherwise said, in a small sample, task-irrelevant facial emotional expressions could drive misleading effects on behavioral responses just by chance. In Berger et al. [37], Mirabella [23], and in the current research, a relatively large number of participants was tested in two tasks using the same images, but changing the task-relevance of emotions. This approach allowed a direct test of the effect of task-relevance and provided a clear outcome.

### 4.2. The Effect of Angry, Fearful, and Happy Faces

It has been commonly stated that threatening but not positive facial expressions elicit rapid action associated with fight–flight behavior [38]. However, we showed evidence that facial expressions of anger and fear significantly decrease response readiness with respect to happy faces. These results are in line with those of Mirabella [23] and Berger et al. [37]. Furthermore, angry faces also interfered with movement execution as they slowed down MTs and decreased accuracy with respect to fearful and happy expressions. Berger et al. [37] suggested that positive expressions improved working memory performance because participants recognize happiness more efficiently than negative facial expressions. Such an argument is supported by evidence indicating that it is easier to detect either static [39,40] or dynamic [41] facial displays of happiness than anger. However, although in our sample, happy faces are recognized better than angry and fearful faces, we did not find any correlations between the recognizability of facial expressions and the corresponding effect on behavioral parameters. In our opinion, the impact of threatening stimuli could be explained in terms of attentional capture. All salient affective stimuli are known to attract attention [2,42]. However, threatening emotions exert a more efficient capture of attentional resources [43,44] than positive emotions. Such hypervigilance could make it hard to direct attention away from threats once detected [45]. This phenomenon is stronger for angry facial expressions than for fearful ones because, while the former indicates the presence of a threat in the surrounding environment, the latter indicates direct threats to the observer, i.e., a more salient message. As such, angry faces are likely to require more accurate screening to uncover others’ intentions by diverting processing away from the ongoing action. The stronger attentional capture by angry stimuli might also explain the effects on MTs. It was previously shown that reaching arm movements are not ballistic, and MTs can be strongly influenced by the participants’ behavioral strategy [46]. This evidence motivated us to check for the effect of facial emotional expression on MTs. We found that angry faces, when task-relevant, increased MTs with respect to all other facial expressions, suggesting that participants’ attention is attracted even during the execution of the reaching arm movement. In the literature, emotional stimuli impact on MTs has rarely been taken into account with contrasting results. On the one hand, two studies showed that negatively valenced stimuli increased MTs [47,48]. On the other hand, Lu et al. [49] did not find an effect of emotional stimuli on MTs. Importantly, in all instances, stimuli valence and arousal were task-irrelevant. Thus, even though more studies are warranted to further identify the effect of emotions on MTs, we demonstrated, for the first time, that negative stimuli strongly capable of attracting attention, such as angry facial expression, can affect MTs.

At first sight, this explanation might seem to contrast with the fact that happy faces are remembered more quickly than threatening emotions. However, this effect might stem from the fact that positive stimuli are rewarding [41], making them more pleasurable to be remembered. Another non-mutually exclusive explanation could be that participants have categorized threatening faces as false alarms during the recall phase, i.e., not behaviorally relevant. Such phenomena could explain why the recall of emotional faces was similar after both the Emotion and Gender discrimination task, i.e., it was not affected by the task-relevance of emotions. Further experiments are needed to clarify this issue.

## 5. Conclusions

Our results indicate that facial emotional expressions must be task-relevant to elicit systematic behavioral effects. Furthermore, we found that angry faces induce quantitatively larger effects with respect to those of fearful faces. We suggest that threatening facial expressions are likely to capture and hold attention more strongly than happy faces to allow people to evaluate potential threats in an observer’s environment. This effect is larger for angry than for fearful faces because the former is potentially more salient as it could indicate a menace directed toward the observer. Thus, angry expressions are likely to require longer scrutiny than fearful expressions. These findings could allow a better understanding of psychiatric and neurological disorders characterized by deep alterations of interpersonal relationships (e.g., autism spectrum disorders and sociopathic personality disorders).

## Figures and Tables

**Figure 1 brainsci-10-00794-f001:**
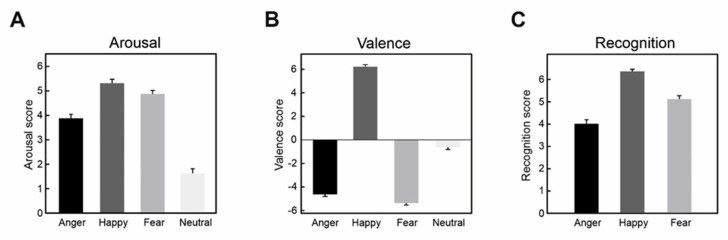
(**A**) Arousal scores for each facial expression. (**B**) Valence scores for all facial expressions. (**C**) Recognition scores of anger, happiness, and fear for the corresponding pictures of angry, happy, and fearful faces. Error bars depict standard error of the means.

**Figure 2 brainsci-10-00794-f002:**
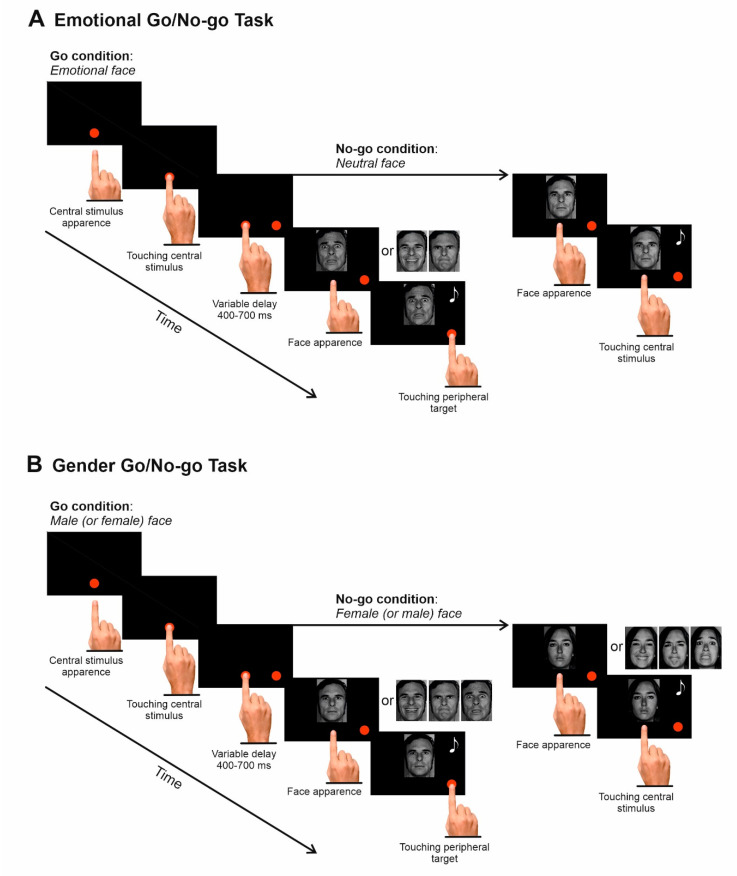
(**A**) Emotion-discrimination task. Each trial started with the presentation of a red circle at the center of the screen. Subjects had to touch it, and, immediately after, a peripheral red circle appeared. After holding the central stimulus for a variable period, it disappeared, and a picture of one of four facial expressions appeared. Participants were instructed to reach and hold the peripheral target when the face expressed an emotion (happiness, fear, or anger. Go condition) or to keep holding the central position when the face displayed a neutral expression (No-go condition). Correct trials were signaled with acoustic feedback (represented in the picture by a musical note). (**B**) Gender discrimination task. The sequence of the events was the same as (**A**). However, in the male version, participants were instructed to reach and hold the peripheral target only when a male face was shown irrespective of the depicted emotion (Go-condition) and to refrain from moving when a female face was presented (No-go condition). Vice versa in the female-version.

**Figure 3 brainsci-10-00794-f003:**
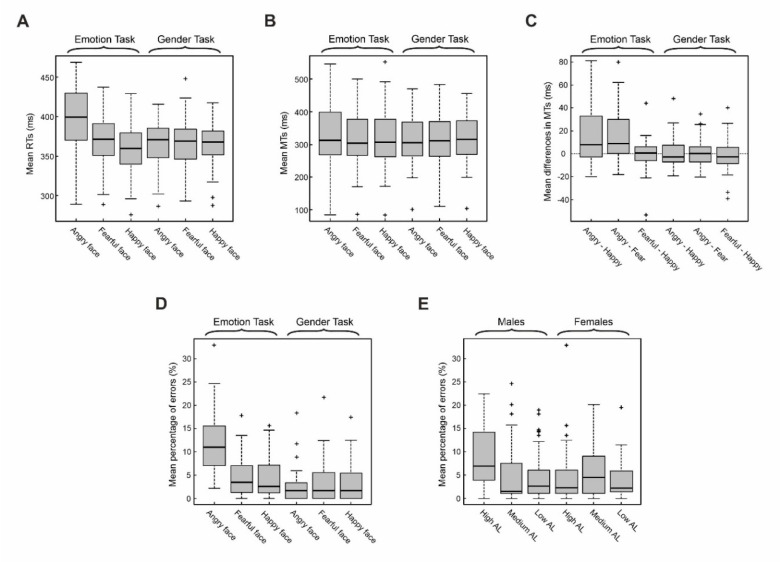
(**A**) Effect of emotional facial expressions on reaction times (RTs). Mean RTs for fearful, happy, and angry faces in the Emotion discrimination task (on the left) and Gender discrimination task (on the right). Only in the Emotion discrimination task, participants were slower when the Go-signal was an angry face than when it was a happy or fearful face (Table 2). (**B**) Effect of emotional facial expression on movement times (MTs). Average MTs in the Emotion discrimination task are displayed on the left and MTs in the gender recognition task on the right. In the Emotion discrimination task, participants were slower when the Go-signal was an angry face than when it was a happy or fearful face (Table 3). (**C**) Mean of the differences of MTs between each pair of emotional facial expressions in each of the two tasks. (**D**) Effect of emotional facial expression on the percentage of errors. Mean percentage of errors to angry, fearful, and happy faces in the Emotion discrimination task (on the left) and Gender discrimination task (on the right). Participants made more mistakes in the Emotion discrimination task than in the Gender discrimination task. In addition, in the Emotion discrimination task, a higher percentage of mistakes occurred when the Go-signal was an angry face than when it was a happy or fearful face (Table 4). (**E**) Effect of arousal level (AL) on the percentage of errors. Results were split according to the participants’ gender. Overall, males made fewer mistakes than females. However, males with high AL made more mistakes than in any other condition (Table 4). In each box plot, the boundary of the box closest to zero indicates the first quartile, a black line within the box marks the median, and the boundary of the box farthest from zero indicates the third quartile. Whiskers indicate values 1.5 times the interquartile range below the first quartile and above the third quartile.

**Figure 4 brainsci-10-00794-f004:**
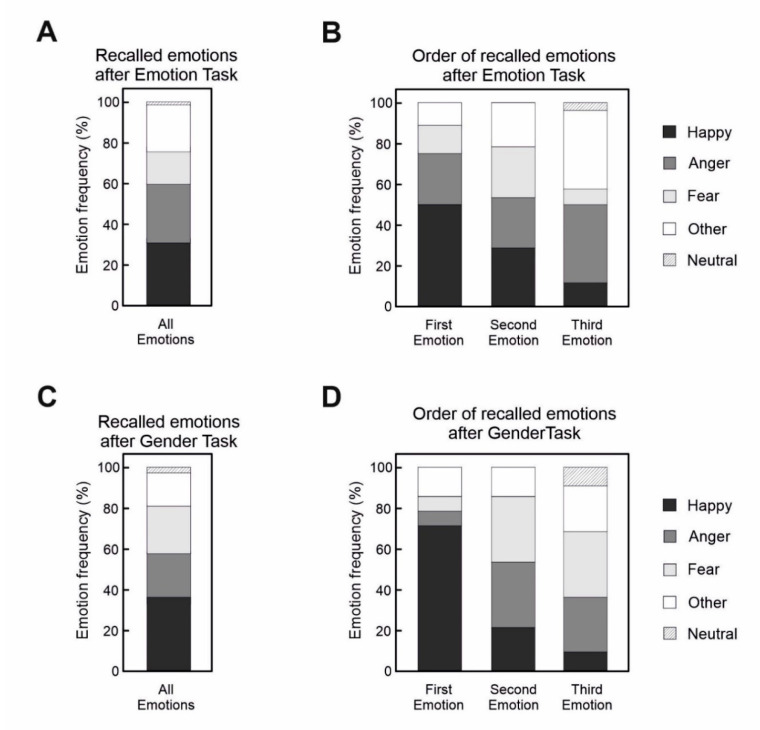
(**A**) Frequencies of recalled emotions after the execution of the Emotion discrimination task. (**B**) Frequencies of recalled emotions after the execution of the Emotion discrimination task according to the recall order. (**C**) Frequencies of recalled emotions after the execution of the Gender discrimination task. (**D**) Frequencies of recalled emotions after the execution of the Gender discrimination task, according to the recall order.

**Table 1 brainsci-10-00794-t001:** Results of the statistical analysis of arousal, valence, and recognizability scores. Post hoc tests (pairwise comparisons) had an adjusted alpha level, according to Bonferroni corrections. Statistically significant results are reported in bold. Bayes factors report the ratio of likelihood of the alternative hypothesis to the likelihood of the null hypothesis (BF_10_), η_p_^2^, partial eta squared, Cohen’s d, ANOVA, analysis of variance.

One-Way ANOVA on Arousal: Emotion (Anger, Happiness, Fear)				
	**Value of Parameters**	***p* values**	**Effect Size**	**BF_10_**
Main effect: Emotion	F(2,110) = 44.8	***p* < 0.0001**	η_p_^2^ = 0.45	9.46 × 10^6^
Post hoc tests:				
Anger vs. Happiness	t(55) = 8.28	***p* < 0.0001**	d = 1.19	1.49 × 10^6^
Anger vs. Fear	t(55) = 8.81	***p* < 0.0001**	d = 0.94	5934
Happiness vs. Fear	t(55) = 2.48	***p* = 0.049**	d = 0.38	1.16
**One-way ANOVA on Valence: Emotion (Anger, Happiness, Fear)**				
	**Value of Parameters**	***p* values**	**Effect Size**	**BF_10_**
Main effect: Emotion	F(2,110) = 1558.0	***p* < 0.0001**	η_p_^2^ = 0.97	1.90 × 10^99^
Post hoc tests:				
Anger vs. Happiness	t(55) = 38.4	***p* < 0.0001**	d = 8.01	9.19 × 10^65^
Anger vs. Fear	t(55) = −4.38	***p* = 0.00016**	d = 0.50	4.27
Happiness vs. Fear	t(55) = 49.4	***p* < 0.0001**	d = 10.5	1.63 × 10^78^
**One-way ANOVA of Recognition scores: Emotion (Anger, Happiness, Fear)**				
	**Value of Parameters**	***p* values**	**Effect Size**	**BF_10_**
Main effect: Emotion	F(2,110) = 96.6	***p* < 0.0001**	η_p_^2^ = 0.64	1.67 × 10^17^
Post hoc tests:				
Anger vs. Happiness	t(55) = 12.5	***p* < 0.0001**	d = 2.07	1.27 × 10^16^
Anger vs. Fear	t(55) = 6.88	***p* < 0.0001**	d = 0.897	2645
Happiness vs. Fear	t(55) = 7.95	***p* < 0.0001**	d = 1.22	3.30 × 10^6^

**Table 2 brainsci-10-00794-t002:** Results of the statistical analysis on reaction times (RTs). Post hoc tests (pairwise comparisons) had an adjusted alpha level corrected according to Bonferroni. Statistically significant results are reported in bold. Bayes factors report the ratio of likelihood of the alternative hypothesis to the likelihood of the null hypothesis (BF_10_). ANOVA, analysis of variance. Measures of size effects: η_p_^2^ for ANOVAs and Cohen’s d for post hoc tests. Differences in the estimated marginal means (M_diff_) are reported along with their 95% confidence interval (CI).

Four-way ANOVA of RTs: Emotion (Anger, Happiness, Fear); Sex (F, M); AL (High, Medium, Low);						
Task (Emotion Discrimination Task, Gender Discrimination Task)						
	**Value of Parameters**	***p* values**	**M_diff_**	**95% CI**	**Effect Size**	**BF_10_**
Main effect: Emotion	F(1.54,76.8) = 51.8	***p* < 0.001**			η_p_^2^ = 0.51	3.1 × 10^7^
Post hoc Tests:						
Anger vs. Happiness	t(111) = 8.46	***p* < 0.001**	19.27	(13.63, 24.92)	d = 0.80	4.4 × 10^8^
Anger vs. Fear	t(111) = 6.80	***p* < 0.001**	15.07	(9.58, 20.55)	d = 0.64	1.2 × 10^6^
Happiness vs. Fear	t(111) = -3.14	***p* = 0.008**	−4.21	(−7.53, −0.89)	d = 0.30	12.20
Main effect: Task	F(1,50) = 5.15	***p* = 0.028**	8.86	(1.02, 16.70)	η_p_^2^ = 0.09	305
Interaction: Emotion*Task	F(1.77,88.5) = 47.10	***p* < 0.001**			η_p_^2^ = 0.49	5.7 × 10^7^
Post hoc Tests:						
Emotion Task-Anger vs. Happiness	t(55) = 10.08	***p* < 0.001**	37.02	(27.92, 46.12)	d = 1.35	1.9 × 10^12^
Emotion Task-Anger vs. Fear	t(55) = 7.35	***p* < 0.001**	27.26	(18.07, 36.45)	d = 0.98	9.3 × 10^7^
Emotion Task-Happiness vs. Fear	t(55) = −5.31	***p* < 0.001**	−9.76	(−14.31, −5.21)	d = 0.71	1.8 × 10^4^
Gender Task-Anger vs. Happiness	t(55) = 0.76	*p* = 1	1.53	(−3.48, 6.53)	d = 0.10	0.20
Gender Task-Anger vs. Fear	t(55) = 1.26	*p* = 0.640	2.87	(−2.75, 8.48)	d = 0.17	0.21
Gender Task-Happiness vs. Fear	t(55) = 0.60	*p* = 1	1.34	(−4.16, 6.85)	d = 0.08	0.15
Emotion Task-Anger vs. Gender Task-Anger	t(55) = 5.92	***p* < 0.001**	28.83	(19.05, 38.61)	d = 0.79	2.6 × 10^5^

**Table 3 brainsci-10-00794-t003:** Results of the statistical analysis of movement times (MTs). Post hoc tests (pairwise comparisons) had an adjusted alpha level corrected according to Bonferroni. Statistically significant results are reported in bold. Bayes factors report the ratio of likelihood of the alternative hypothesis to the likelihood of the null hypothesis (BF_10_). ANOVA, analysis of variance. Measures of size effects: η_p_^2^ for ANOVAs and Cohen’s d for post hoc tests. Differences of the estimated marginal means (M_diff_) are reported along their 95% confidence interval (CI).

Four-way ANOVA of MTs: Emotion (Anger, Happiness, Fear), Sex (F, M), AL (High, Medium, Low),						
Task (Emotion Discrimination Task, Gender Discrimination Task)						
	**Value of Parameters**	***p* values**	**M_diff_**	**95% CI**	**Effect Size**	**BF_10_**
Main effect: Emotion	F(1.55,77.3) = 16.64	***p* < 0.001**			η_p_^2^ = 0.25	2.82
Post hoc Tests:						
Anger vs. Happiness	t(111) = 4.18	***p* < 0.001**	8.10	(3.31, 12.90)	d = 0.40	431
Anger vs. Fear	t(111) = 4.75	***p* < 0.001**	8.31	(3.98, 12.65)	d = 0.45	3120
Happiness vs. Fear	t(111) = 0.19	*p* = 1	0.21	(−2.61, 3.03)	d = 0.02	0.12
Interaction: Emotion vs. Task	F(1.99,99.5) = 16.54	***p* < 0.001**			η_p_^2^ = 0.25	1.49
Post hoc Tests:						
Emotion Task-Anger vs. Happiness	t(55) = 4.84	***p* < 0.001**	14.59	(7.12, 22.05)	d = 0.65	3208
Emotion Task-Anger vs. Fear	t(55) = 5.63	***p* < 0.001**	15.82	(8.85, 22.78)	d = 0.75	2.0 × 10^4^
Gender Task-Anger vs. Happiness	t(55) = 0.98	*p* = 0.99	1.62	(−2.47, 5.71)	d = 0.13	0.16
Gender Task-Anger vs. Fear	t(55) = 0.55	*p* = 1	0.81	(−2.80, 4.42)	d = 0.07	0.17
Gender Task-Happiness vs. Fear	t(55) = −0.44	*p* = 1	-0.81	(−5.38, 3.76)	d = 0.06	0.15
Emotion Task-Anger vs. Gender Task - Anger	t(55) = 3.17	***p* = 0.003**	18.51	(6.79, 30.23)	d = 0.42	5.67

**Table 4 brainsci-10-00794-t004:** Results of the statistical analysis on the percentage of mistakes. Post hoc tests (pairwise comparisons) had an adjusted alpha level corrected according to Bonferroni. Statistically significant results are reported in bold. Bayes factors report the ratio of likelihood of the alternative hypothesis to the likelihood of the null hypothesis (BF_10_). ANOVA, analysis of variance. Measures of size effects: η_p_^2^ for ANOVAs and Cohen’s d for post hoc tests. Differences of the estimated marginal means (M_diff_) are reported along their 95% confidence interval (CI). AL, arousal level.

Four-way ANOVA of Mistakes: Emotion (Anger, Happiness, Fear, Sex (F, M), AL (High, Medium, Low),						
Task (Emotion Discrimination Task, Gender Discrimination Task)						
	**Value of Parameters**	***p* values**	**M_diff_**	**95% CI**	**Effect Size**	**BF_10_**
Main effect: Emotion	F(1.69, 84.8) = 37.44	***p* < 0.001**			η_p_^2^ = 0.43	9.9 × 10^5^
Post hoc Tests:						
Anger vs. Happiness	t(111) = 7.01	***p* < 0.001**	3.54	(2.29, 4.79)	d = 0.66	4.9 × 10^4^
Anger vs. Fear	t(111) = 6.56	***p* < 0.001**	3.27	(2.03, 4.50)	d = 0.62	1.0 × 10^5^
Happiness vs. Fear	t(111) = −0.79	*p* = 1	−0.27	(−1.13, 0.58)	d = 0.07	0.11
Main effect: Task	F(1, 50) = 44.54	***p* < 0.001**	3.81	(2.66, 4.96)	η_p_^2^ = 0.47	5.8 × 10^13^
Main effect: Sex	F(1, 50) = 5.27	***p* = 0.026**	1.60	(0.20, 3.01)	η_p_^2^ = 0.10	0.44
Main effect: AL	F(2, 50) = 3.97	***p* = 0.025**			η_p_^2^ = 0.14	0.19
Post hoc test: High vs. Low	t(34) = 2.81	***p* = 0.021**	2.45	(0.29, 4.61)	d = 0.47	1.42
Interaction: Emotion*Task	F(1.56, 78.21) = 47.64	***p* < 0.001**			η_p_^2^ = 0.49	8.3 × 10^13^
Post hoc Tests:						
Emotion Task-Anger vs. Happiness	t(55) = 7.58	***p* < 0.001**	7.55	(5.09, 10.02)	d = 0.72	9.7 × 10^7^
Emotion Task-Anger vs. Fear	t(55) = 7.71	***p* < 0.001**	7.03	(4.77, 9.29)	d = 0.73	2.6 × 10^8^
Gender Task-Anger vs. Happiness	t(55) = 1.43	*p* = 0.477	−0.47	(1.30, 0.35)	d = 0.19	0.65
Gender Task-Anger vs. Fear	t(55) = 1.38	*p* = 0.521	−0.49	(1.38, 0.39)	d = 0.18	0.32
Gender Task-Happiness vs. Fear	t(55) = 0.05	*p* = 1	−0.02	(−1.0, 0.96)	d = 0.01	0.16
Emotion Task-Anger vs. Gender Task-Anger	t(55) = 9.80	***p* < 0.001**	9.00	(7.15, 10.84)	d = 1.31	5.8 × 10^11^
Emotion Task-Fear vs. Gender Task-Fear	t(55) = 2.23	***p* = 0.030**	1.48	(0.15, 2.80)	d = 0.30	3.41
Interaction: Sex*AL	F(2, 50) = 5.85	***p* = 0.005**			η_p_^2^ = 0.19	5.97
Post hoc Tests:						
Female-High vs. Male-High	t(50) = 3.83	***p* < 0.001**	4.84	(2.30, 7.39)	d = 0.51	406
Male-High vs. Medium	t(50) = 3.41	***p* = 0.004**	4.31	(1.17, 7.45)	d = 0.45	47.10
Male-High vs. Low	t(50) = 3.38	***p* = 0.004**	4.42	(1.18, 7.65)	d = 0.45	80.30

**Table 5 brainsci-10-00794-t005:** Results of the Chi-squared goodness-of-fit test (χ^2^) of recalled emotion frequencies in the first, second, and third place of recalling. Standardized residuals (SR) were used to compute *p*-values for the post hoc tests. Statistically significant results are reported in bold.

	Recalled Emotions after Emotion Discrimination Task		Recalled Emotions after Gender Discrimination Task	
	**Value of Parameters**	***p* Values**	**Value of Parameters**	***p* Values**
Total recalled emotions:	χ^2^ (2, N = 62) = 4.29	*p* = 0.117	χ^2^ (2, N = 63) = 3.52	*p* = 0.17
First recalled emotion:	χ^2^ (2, N = 25) = 6.32	***p* = 0.042**	χ^2^ (2, N = 24) = 27.0	***p* < 0.0001**
Post hoc: Happiness	SR = 2.40	***p* = 0.049**	SR = 5.20	***p* < 0.0001**
Post hoc: Anger	SR = -0.57	*p* = 1	SR = −2.60	***p* = 0.028**
Post hoc: Fear	SR = −1.84	*p* = 0.20	SR = −2.60	***p* = 0.028**
Second recalled emotion:	χ^2^ (2, N = 22) = 0.09	*p* = 0.956	χ^2^ (2, N = 24) = 0.75	*p* = 0.687
Third recalled emotion:	χ^2^ (2, N = 15) = 7.60	***p* = 0.022**	χ^2^ (2, N = 15) = 2.80	*p* = 0.247
Post hoc: Happiness	SR = −1.10	*p* = 0.82	-	-
Post hoc: Anger	SR = 2.74	***p* = 0.019**	-	-
Post hoc: Fear	SR = −1.64	*p* = 0.30	-	-

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
