# Peer review of "Threatening Facial Expressions Impact Goal-Directed Actions Only if Task-Relevant"

_brainsci, 2020, doi:10.3390/brainsci10110794_

Round 1
Reviewer 1 Report
I thank the authors for their point-by-point response to the comments we gave them. I have no issue with the minor points and also generally accept their rebuttals regarding the more major points except the below.
Major point 4: The authors’ reply still requires a clarification. Specifically, what does it mean to "abort a trial"? If the participants were left to finish the task if the RT was between 500 and 600 ms and at the same time, their RT was recorded, then what exactly was the effect of abortion?
Major point 6: The authors' solution of providing the null results in Supplementary tables sounds fine. However, the sentence below in the Discussion still needs to be supported by a null effect reported in the Results section:
"Our evidence indicates that valence but not arousal or recognizability of the stimuli affects the generation and execution of actions when emotional facial expressions are task-relevant, i.e., during the Emotion discrimination task."
This statement references a null effect of the interaction between Task and Arousal, which is not reported in the results.
Major point 8: I am happy to acknowledge my previous misunderstanding regarding the data structure. Though I would urge the authors to read their text again to make sure other readers will not repeat my mistake. What is more, it is still not completely clear how the supported null effect of Emotion in the Gender discrimination task favors the idea that there is no sampling bias. There may well not be oneI do not claim that there is a sampling bias, I just do not understand the line of reasoning behind this argument.
Author Response
Please, see the attached file.

Reviewer 2 Report
The authors made a substantial effort to improve the manuscript. It is now clear, why another experiment regarding anger emotion is needed beside the existing Mirabella 2018 and the authors are not repeating themselves by just squeezing some more results from the same experiment. However, a discussion focusing on the interaction of anger and movement is missing. The current discussion is mostly drifted into the general interaction of emotion and motion. I suggest the authors expand the discussion to cover this central findings of the experiment.
Author Response
Please, see the attached file.

This manuscript is a resubmission of an earlier submission. The following is a list of the peer review reports and author responses from that submission.
Round 1
Reviewer 1 Report
The authors present an overall compelling argument that emotional stimuli influence motor control only if task relevant. While the argument made is clear and the experiment supporting it carefully conducted, there are several points, both major and minor, which should be addressed by the authors.
Major points:
- Conceptually, it is debatable whether the emotional expression is indeed completely task-relevant in the Emotion discrimination task. Participants were not asked to identify which emotion they saw in the expression; instead, they were asked to decide whether there was or was not an emotion. While the Emotion discrimination task instruction clearly drew their attention towards the emotional expression, it still required them just to decide whether they saw an emotion. That is a different task and likely simpler than to decide what emotion it is. It would be good to see this brought up in the introduction and discussed in the discussion (section 4).
- The monitor used (640 × 480 px by 17 inches) was of rather low resolution, which might hinder more subtle emotion recognition. This might artificially amplify the effect observed (and therefore limit the study’s ecological validity). Can the authors speak to the use of this monitor in emotion-recognition experiments?
- The description of the Emotion Discrimination task (lines 143–152) leaves the specific instructions that the participants received unclear. It is later implied that they were not told that there will be three emotions and one neutral stimulus, because they were asked about it within the Emotion Recall task. So, how exactly were the participants instructed to react to the emotional stimuli? What was the specific wording? Were they told that some images will be emotionless?
- Regarding lines 165–172:
- If any trial in which the RT was about to be longer than 500 ms was aborted, how could these trials be included in the analyses? Should they not be classified as errors? This is confusing and would be good to clarify.
- If these overtime reaching trials were included in the analyses, as the text suggestion, what were their RT values? If I understand correctly, since the trials were aborted, the RT was taken as 500 ms, because the authors state that this is to avoid cutting the RT distribution tail. But, on the contrary, this procedure would effectively cut the tail by replacing all values higher than 500 by exactly 500.
- During the data analysis it appears as though the authors may not have accounted for the random effect of individual subjects. They discuss the interindividual variability later; but, despite that, if the individual trials were considered as individual data points in the analyses without considering the effect of individual participants, the ANOVA's assumption of independent observations is violated. If that is the case, this problem could be overcome using a mixed-effect linear model.
- Tables 3, 4 and 5 appear to be missing information on some of the factors, which were included in the model (implied by the lines 210–214). One might assume that these are null results, but those should still be reported, especially if a Bayes-factor analysis is employed and null results are discussed in the Discussion (see the line 355–356). Specifically, Arousal and Sex results are missing from Tables 3 and 4. Also, Tables 3 and 4 miss some combinations of the post-hoc tests without any explanation (there is only one cross-task interaction for Anger, but none for Fear or Happiness). Table 5 misses these more complex comparisons completely.
- The authors state that "Given that we compared the responses elicited by the same stimuli across the same population in the two tasks, they are unlikely to depend on the subjects’ variability or the stimuli features." (lines 363–365). This is not very convincing because possible effects of subjects' individuality was not ruled out (especially for the between-subject effect of Arousal) and should be considered in the analyses. Could the authors comment on this?
- While it is true that BF values favor the null hypothesis in the Gender discrimination task (and hence suggest that the test was not underpowered), it is not clear how this prove that there is no sampling bias (lines 379–381). A clarification to this argument is welcome.
- The authors note "Single subjects can show an impact of emotion, even though they are task-irrelevant. Therefore, especially if the sample is relatively small, a random effect could be observed." in the context of previous studies (lines 384–386). They therefore appear to admit possible effect of sample variability. Yet elsewhere (see major points 7 and 8) they insist that the sample variability should not play a role.
- In subsection 4.2, the authors address the possibility that longer RTs and MTs and more errors on negative emotions and especially anger might be caused by variability in recognizability. At the same time, the authors have data on the recognizability ratings of the images. So, the logical step would be to perform the analysis on the relation between recognizability ratings on one side and RTs, MTs and errors on the other. This could be elegantly done by including the recognizability ratings into the mixed-effect model proposed above.
- A general impression from the Discussion section (and especially subsection 4.2) is that the authors overstate the similarity of the effects of angry vs. fearful faces. There were significant differences between these two emotions in all three studied variables (RTs, MTs and Error rates). And this should be further reflected in the discussion and reasons for the difference should be suggested.
- There is a large number of minor comments below. Each one of them seems addressable. But the overall impression is unfortunately that the authors could have done more to submit a more thoroughly revised manuscript.
Minor points:
- There are several instances where virtually the same information is duplicated in different forms and places. This is especially relevant to Table 1, Figure 1 and lines 116–125 of the text, which all present virtually the same data. Similar duplication can be found throughout the Results section. This is redundant.
- It is stated on lines 140–141 that order of the tasks administered was counterbalanced. However, was the same was true for the order of the actual images? Were they shuffled, systematically counterbalanced, or fixed?
- There is a slight confusion regarding Figure 2. On lines 147 and 148, the authors note that the middle red circle disappeared simultaneously with the image presentation. However, Figure 2 specifically shows the middle red circle and the image displayed in the same frame (see penultimate frame in every condition). Should the image should be corrected?
- In the sentence on lines 149–150 the authors might want to clarify what reaching the peripheral target means. Does the participant drag the finger over the screen or lifts it up and touches on the peripheral circle? It is implied in section 2.7 but might be more useful if explained in the task description.
- Line 152 states that “Acoustic feedback indicated successful trials.” But what exactly qualifies as a successful trial?
- Line 171 leaves unclear whether 25 is a reference to a source in the literature.
- The following sentence (lines 178–181) is confusing: “In the male control condition, each picture showing a male face was presented 132 times (Go-trials, frequency: 67%), whereas each female face was presented 66 times (No-go trials, frequency: 33%), and vice versa for the female control condition. Overall, participants had to perform 396 trials.” There were 8 pictures of male faces and 8 pictures of female faces. Therefore, if "each picture showing a male face was presented 132 times", that would make 8 × 132 = 1056 images, which is already above the 369 total trials. Might this be resolved by rewording of the phrase “each picture”?
- It is not stated in the subsection 2.5 whether the overtime-reaching trials in the Gender Discrimination task were included in the analyses, as were these trials in the Emotion Discrimination task.
- On lines 192–193, the sentence “Given this, we considered just the first three recalled emotions.” needs clarification. Considered for what?
- Participants were subdivided into three categories based on their arousal level (lines 205–209). What exactly were the criteria for the categorization (i.e. what were the thresholds?) and how many participants ended up in which category?
- For the sake of being completely precise when discussing the effect size measures (lines 215–218), perhaps clarify that values ABOVE the stated values are considered to indicate large, medium and small effects (i.e. that these values are considered thresholds).
- The wording "moved more slowly" on line 240 is ambiguous. It seems like it refers to movement time (MT) while it more likely refers to reaction time (RT).
- Some columns in Tables 3 and 5 are not wide enough for the presented data (perhaps a formatting issue).
- Values in Tables 3, 4, and 5 are presented with inconsistent number of decimals. A consistent presentation of numbers makes reading easier.
- Why is Emotion Recall Performance placed in a dedicated subsection 3.1, while the RT, MT and Error analyses are not? It might make the text more accessible if these were in a dedicated subsection or subsections as well.
- On lines 359–360, the authors again talk about "slowing down" a movement, while it seems that they are referring to RTs, not MTs.
- In subsection 4.2, the authors suddenly switched to APA citation format. Furthermore, Schutter et al. (2008), Goren & Wilson (2006), Melcher et al. (2012), Becker et al. (2006), and Becker et al. (2012) are not listed in the reference list.
Reviewer 2 Report
Mancini et al investigated the interaction of facial affective information (as encoded from facial emotion expressions) with action planning using a paradigm already published by one of the authors (Mirabella 2018).
The hypotheses and findings contribute to understanding the interaction between facial emotions and action with an implication in the field of social neuroscience, which makes it a notable research.
The main problem I see, is that there might not be enough new results and interpretations that justify a stand alone paper next to Mirabella 2018. One suggestion might be to link the neuroimaging findings (and there are plenty of) to the current results to add to the available knowledge about emotion-related motor processing.
The authors explain the motivation of the study, which is a replication of an already published study as follows: “whether the effect of negative emotions is always the same”. I was surprised that again one positive emotion and only 2 various negative emotions has been picked up and not more than 2 negative emotions (for example sadness and disgust according to Eckman). Please provide a plausible reason choosing the emotions.
Generally, the manuscript needs to have a unified nomenclature regarding all various terms like Motor readiness (title) /motor response/motor control (keywords)/movement times/etc. If motor planning and execution are differentiable based on the design, an explanation is needed.
The abstract needs a total revision to provide a meaningful summary of the design and the results, highlighting the main findings in a structured way. Please avoid over interpretation of the results in the abstract.
What is the hypothesis behind the emotion recall test?
The experimental and statistical methods applied to this study are sound. There are a few questions, that need clarifications:
Why only male subjects included? Please provide reasons for this exclusion of females. How are the results can be generalized to the female gender? Please discuss.
How was the instruction provided to the subject? Was that a structured and unified instruction? Did all participants get that their output would be evaluated based on RT und performance both? (a fast and as correct as possible?)
Results:
Figure3c: in a factorial design, subtractions of factors do not seem to be statistically relevant.
Comments regarding discussion:
Line 348: “facial expressions of anger and fear are often generically considered as threatening stimuli”. I am not sure if this assumption is common in the literature. Please provide references.
Line 350: “arousal of the stimuli is still rarely considered”. Again, I don’t believe that this claim is true. There is a huge body of literature considering arousal as a critical factor. Please provide e reference or modify your argument.
Line 420: how do you come to this conclusion? Automaticity was not tested only RT and accuracy.
